# Using Artificial Intelligence as a Risk Prediction Model in Patients with Equivocal Multiparametric Prostate MRI Findings

**DOI:** 10.3390/cancers18010028

**Published:** 2025-12-21

**Authors:** Abdullah Al-Khanaty, David Hennes, Arjun Guduguntla, Pablo Guerrero, Carlos Delgado, Eoin Dinneen, Elio Mazzone, Sree Appu, Damien Bolton, Renu S. Eapen, Declan G. Murphy, Nathan Lawrentschuk, Marlon L. Perera

**Affiliations:** 1Division of Cancer Surgery, Peter MacCallum Cancer Centre, Melbourne, VIC 3000, Australiapguerreros@clinicasantamaria.cl (P.G.);; 2Department of Surgery, University of Melbourne, Austin Health, Melbourne, VIC 3010, Australia; 3Hudson Institute of Medical Research, Monash University, Clayton, VIC 3168, Australia; 4Division of Oncology/Unit of Urology, Gianfranco Soldera Prostate Cancer Laboratory, Urological Research Institute, IRCCS San Raffaele Scientific Institute, 20162 Milan, Italy; 5Department of Urology, Cabrini Clinical School, Melbourne, VIC 3186, Australia; 6Sir Peter MacCallum Department of Oncology, University of Melbourne, Melbourne, VIC 3000, Australia; 7Department of Surgery, University of Melbourne, Royal Melbourne Hospital, Melbourne, VIC 3052, Australia; 8EJ Whitten Prostate Cancer Research Centre at Epworth, Melbourne, VIC 3121, Australia

**Keywords:** prostate cancer, PI-RADS 3, multiparametric MRI, artificial intelligence, radiomics, machine learning, deep learning

## Abstract

Magnetic resonance imaging of the prostate is widely used to detect prostate cancer, but one common result—called a PI-RADS 3 lesion—remains difficult to interpret. These findings are uncertain: most are not dangerous, yet some contain clinically important cancer. This uncertainty leads to many men undergoing unnecessary biopsies, while others may experience delayed diagnosis of aggressive disease. New artificial intelligence techniques can analyse prostate scans in more detail than the human eye and combine imaging patterns with clinical information to estimate cancer risk more accurately. This review explains why PI-RADS 3 lesions remain a problem, summarises how artificial intelligence has been studied to improve decision-making in this setting, and discusses how these tools could be safely integrated into routine care. If validated and implemented carefully, artificial intelligence may help doctors reduce unnecessary procedures, improve cancer detection, and provide more consistent care for patients with uncertain prostate scan findings.

## 1. Introduction

Prostate cancer is the most common malignancy affecting men worldwide, with a lifetime risk approaching 37 percent in some populations [1,2]. Global incidence is projected to rise substantially over the coming decades; by the year 2040, prostate cancer is expected to account for approximately 2.3 million new diagnoses and more than 740,000 deaths annually [3]. Against this epidemiological backdrop, there is a pressing need for diagnostic pathways that can accurately identify clinically significant prostate cancer (csPCa) while minimising the harms of overdiagnosis, overtreatment, and unnecessary biopsy [4]. 

Multiparametric prostate magnetic resonance imaging (mpMRI) has become the central imaging tool for the detection, localisation, and risk stratification of prostate cancer [5]. mpMRI integrates anatomical and functional sequences—high-resolution T2-weighted imaging for zonal anatomy, diffusion-weighted imaging (DWI) and apparent diffusion coefficient (ADC) mapping for assessing tissue cellularity, and, where used, dynamic contrast enhancement (DCE) for characterising vascularity [6,7,8]. The combination of these sequences enhances the differentiation between csPCa and benign mimics and has been incorporated into major international guidelines for both biopsy-naïve and previously biopsied men [9,10].

The practice-changing PROMIS and PRECISION trials established the superiority of MRI-directed diagnostic pathways. PROMIS demonstrated that mpMRI has substantially greater sensitivity for detecting csPCa than transrectal ultrasound (TRUS) biopsy alone and can safely reduce the need for immediate biopsy in many men [9,10]. PRECISION subsequently showed that an MRI–targeted biopsy pathway improves detection of csPCa while reducing the diagnosis of indolent disease [10]. These findings underpin current global recommendations that endorse mpMRI as the first-line investigation in men with suspected prostate cancer and highlight its central role across diagnosis, targeted biopsy, active surveillance selection, and treatment planning [11].

The Prostate Imaging Reporting and Data System (PI-RADS) version 2.1 standardised mpMRI acquisition and interpretation, offering a widely adopted five-point scale describing the probability of csPCa [5]. PI-RADS 1 and 2 lesions have very low csPCa likelihood and are typically observed, whereas PI-RADS 4 and 5 lesions carry high suspicion and warrant targeted biopsy [12]. However, PI-RADS 3 lesions occupy an intermediate and inherently uncertain category. They represent an equivocal probability of csPCa and do not clearly satisfy criteria for benignity or high suspicion. Their management is not standardised across guidelines or institutions, leading to considerable variation in clinical practice.

These interpretive limitations have intensified interest in the application of artificial intelligence (AI). Radiomics, machine learning (ML), and deep learning (DL) methods can identify imaging features beyond human perception, quantify tissue heterogeneity, and integrate imaging with clinical parameters to generate individualised risk predictions [11]. The recent PI-CAI international benchmark demonstrated that state-of-the-art AI systems can match or exceed the diagnostic performance of 62 expert radiologists across multiple countries and MRI vendors [13]. These findings highlight AI’s potential to enhance reproducibility, reduce inter-reader variability, and support more consistent decision-making.

The aim of this review is to synthesise current evidence regarding AI-enabled risk prediction in men with equivocal mpMRI findings, with particular emphasis on PI-RADS 3 lesions. We examine methodological approaches, summarise diagnostic performance across radiomics, ML, and DL models, identify limitations within the evidence base, and outline key priorities for clinical translation and implementation.

## 2. The Clinical Problem of PI-RADS 3

PI-RADS 3 lesions sit at the centre of diagnostic uncertainty in prostate MRI interpretation. By definition, PI-RADS 3 indicates an “equivocal likelihood of clinically significant prostate cancer,” reflecting a level of suspicion insufficient for upgrading to PI-RADS 4 but not low enough to confidently assign PI-RADS 2 [5,14]. Because the criteria for PI-RADS 3 differ by anatomical zone, the category encompasses a broad spectrum of imaging appearances, contributing to its heterogeneity and interpretive difficulty.

In the peripheral zone, where DWI is the dominant determining sequence, PI-RADS 3 lesions are characterised by mild to moderate hypointensity on ADC maps and iso- to mildly hyperintense signal on high b-value DWI [14]. T2-weighted imaging typically demonstrates heterogeneous or mildly hypointense signal without well-defined margins. Importantly, the lesion must lack features that meet criteria for PI-RADS 2 (benign), PI-RADS 4, or PI-RADS 5 (highly suspicious). The absence of definite focal early enhancement on DCE is a required feature; the presence of focal enhancement would otherwise upgrade the lesion to PI-RADS 4 [14].

In the transition zone, where T2-weighted imaging is dominant, PI-RADS 3 lesions exhibit heterogeneous signal intensity and obscured or ill-defined margins, distinguishing them from the well-circumscribed, encapsulated nodules typical of PI-RADS 2. These lesions may show mild to moderate ADC hypointensity and iso- to mildly hyperintense signal on high b-value DWI. Unlike in the peripheral zone, focal enhancement on DCE may be present, but this alone does not warrant upgrading to PI-RADS 4 [14]. Transition-zone PI-RADS 3 lesions are particularly challenging because benign prostatic hyperplasia and stromal nodules often mimic csPCa [5].

PI-RADS 3 lesions occur in approximately 22 to 32 percent of mpMRI examinations, making them one of the most frequent reasons for clinical uncertainty [4]. In two large cohorts of men with mixed first and previously negative biopsies, the prevalence of maximal PI-RADS 3 score was 31% (196/625) [15] and 32% (367/1159) [16]. Their prevalence is expected to rise as biparametric MRI (bpMRI) becomes more widely adopted and dynamic contrast enhancement is omitted, removing one of the mechanisms for upgrading peripheral PI-RADS 3 lesions [17,18,19].

Cancer detection rates are heterogeneous. Clinically significant prostate cancer is identified in 10 to 30 percent of PI-RADS 3 lesions [20], with meta-analytic estimates indicating a pooled csPCa detection rate of approximately 25 percent (95 percent confidence interval 18–32 percent) [21]. Although this rate is lower than for PI-RADS 4 and 5 lesions, it is non-trivial, meaning that PI-RADS 3 cannot be dismissed as benign. Even targeted biopsy misses approximately 6 percent of csPCa in PI-RADS 3 lesions [21,22]. This equivocal risk profile generates a dilemma. Biopsying all PI-RADS 3 lesions increases the detection of indolent tumours and exposes men to procedural risks such as sepsis, bleeding, and urinary retention, as well as the psychological burden of overdiagnosis [23]. Conversely, deferring biopsy risks delayed diagnosis of aggressive cancers, potentially reducing opportunities for curative treatment.

A major contributor to the variability in outcomes is the inconsistency in PI-RADS 3 assignment. Inter-reader agreement for PI-RADS v2 interpretations is moderate overall (κ ≈ 0.63), but falls to only fair in the transition zone (κ ≈ 0.53), reflecting the difficulty of distinguishing TZ tumour from benign stromal hyperplasia [24]. This subjectivity leads to substantial variation across centres, radiologists, and clinical teams in how PI-RADS 3 lesions are interpreted and managed.

Major international guidelines from the European Association of Urology and the American Urological Association acknowledge the diagnostic uncertainty of PI-RADS 3 lesions but do not provide a definitive management algorithm. Instead, clinicians are advised to incorporate PSA density, clinical history, and patient preference into decision-making [25]. These adjuncts, however, lack universal validation and contribute to heterogeneity in practice. Together, the prevalence, interpretive ambiguity, limited reproducibility, and guideline variability surrounding PI-RADS 3 highlight an unmet need for objective, reproducible risk-prediction tools. Artificial intelligence is uniquely positioned to address this need by quantifying subtle imaging features, improving inter-reader consistency, and supporting personalised biopsy and surveillance decisions [21,24,26].

## 3. Current Adjuncts in Clinical Practice

Several adjunctive strategies are currently employed to refine biopsy decisions for men with PI-RADS 3 lesions. PSA density (PSAD) remains the most widely used adjunct metric, with thresholds around 0.15–0.20 ng/mL^2^ shown to improve discrimination between benign and malignant lesions; however, its diagnostic performance is inconsistent across populations and varies with prostate volume [27,28]. Clinical risk calculators, such as the University of California Prostate Cancer Risk Calculator (UCLA-PCRC), incorporate age, ethnicity, PSA, presence of absence of abnormal digital rectal examination, prostate volume, and biopsy history to provide individualised risk estimates of clinically significant prostate cancer [29]. While these tools are useful, their calibration often requires localisation, and they are not specifically tailored to equivocal mpMRI findings. Molecular and urine-based biomarkers, including the Prostate Health Index (PHI) [30], 4Kscore, and SelectMDx, offer further avenues for risk stratification. Yet their availability is variable, they add cost, and none have been prospectively calibrated or validated as standard decision aids in the management of PI-RADS 3 lesions. Importantly, the ongoing PRIMARY2 trial is specifically investigating whether adding PSMA PET to mpMRI improves risk stratification in men with PI-RADS 3 lesions; its results are awaited and may inform future guideline-directed management.

## 4. Artificial Intelligence in Prostate Imaging

The application of AI to prostate MRI has rapidly advanced over the past decade. In the context of PI-RADS 3 lesions, where radiologists face diagnostic uncertainty and variable reproducibility, AI offers the promise of objective, reproducible, and scalable decision support. At its core, AI methods can be broadly categorised into purely radiomics and image-based AI and those that use radiomics and image-based AI combined with clinical predictive models.

## 5. Radiomics and Image-Based AI

Radiomics converts prostate MRI into high-dimensional, quantitative descriptors of tissue intensity, texture, shape, and heterogeneity [31]. Features are typically extracted from T2-weighted images, DWI, ADC maps, and sometimes DCE. A standard pipeline involves lesion segmentation (manual, semi-automatic, or automatic), feature extraction, feature selection (e.g., mRMR, LASSO) to reduce redundancy, and modelling with statistical or machine-learning classifiers to predict outcomes such as csPCa. By capturing patterns that are often imperceptible to the human eye, radiomics offers an objective complement to visual PI-RADS assessment—particularly valuable in the equivocal PI-RADS 3 category, where reader variability and transition-zone ambiguity are common [31,32,33].

In a single-centre study of 46 PI-RADS 3 lesions [34], texture features from T2W and ADC imaging could stratify malignancy risk with area under the receiver operating characteristic curves (AUROCs)—which quantify how well a model separates cancers from benign lesions across all possible decision thresholds—of approximately 0.77–0.82 for PCa and csPCa, providing early evidence that quantitative imaging adds diagnostic signal beyond visual reads. Building on this, Hou et al. [35] evaluated 263 men and developed integrated T2W + DWI + ADC radiomics “redefining scores,” achieving AUC 0.89 and outperforming radiologists (inter-reader κ = 0.435), suggesting potential to mitigate variability in PI-RADS 3 interpretation.

Hectors et al. [36] trained a random forest on 107 T2W features in 240 men with PI-RADS 3 index lesions: test-set AUC 0.76, superior to PSA density and prostate volume (AUCs ~0.61–0.62). Brancato et al. [37] analysed PI-RADS 3 and “upgraded PI-RADS 4” lesions using T2/ADC/DCE features, reporting AUC ~0.80–0.89 and underscoring the dominant contribution of T2/ADC. Not all series were strongly positive: in a multicentre cohort (*n* = 158), Lim et al. [38] found only moderate accuracy (AUC 0.64–0.68), highlighting challenges with generalisability outside a single site. Using ADC-based features and SVMs in 155 PI-RADS 3 lesions, Gaudiano et al. [39] reported sensitivity 78% and specificity 76% on a hold-out test set, improving on unaided reads. In a four-centre cohort (*n* = 463), Jin et al. [40] compared per-sequence models (T2W, DWI, and ADC) with an integrated model; the integrated approach reached mean AUC ~0.80 for csPCa with near-identical internal vs. external performance (0.804 vs. 0.801), supporting cross-site generalisability when pipelines are harmonised.

Gravina et al. [41] evaluated four classifiers in 109 men and found that random forest achieved the best performance, with an AUC of 0.83, sensitivity of 81.7%, and specificity of 71.0%, outperforming historical biopsy detection rates. Hectors et al. [36] evaluated 240 men with PI-RADS 3 index lesions and trained a random forest classifier using 107 T2-weighted radiomic features, achieving an AUC of 0.76 in the test set compared with 0.61 for PSA density and 0.62 for prostate volume, underscoring the superiority of radiomics-based ML models over conventional clinical metrics. More recently, Altinaş et al. [42] studied 235 men with PI-RADS 3 lesions, incorporating clinical, imaging, and systemic inflammatory markers into six ML algorithms. Random forest again demonstrated the highest performance, with an accuracy of 0.86, F1 score of 0.91, and AUC of 0.92. Shapley Additive Explanation (SHAP) analysis highlighted tumour ADC, ADC ratio, and PSA density as the strongest predictors of malignancy, with inflammatory indices such as the systemic inflammatory index and neutrophil-to-lymphocyte ratio contributing more than total PSA or age. Zhao et al. [43] advanced the field by applying an XGBoost model in transition-zone PI-RADS 3 lesions, integrating PSA density, ADC-derived radiomic features, and clinical parameters to achieve an AUC of 0.91, outperforming PSAD or PI-RADS scoring alone. Similarly, Lu et al. [44] used biparametric MRI radiomics with LASSO-based feature selection and logistic regression modelling in 233 men, showing that a combined radiomics + PSAD model significantly improved diagnostic accuracy (validation AUC 0.856) over either parameter individually.

Moving beyond single-centre experiences, [45] developed random forest radiomics models across four institutions, reporting AUCs of 0.87–0.89 in both internal and external validation cohorts. The integration of radiomic features with PI-RADS scores improved specificity without reducing sensitivity, reducing false positives in equivocal lesions.

Deep learning (DL) has gained traction as a strategy to overcome the limitations of traditional radiomics and conventional machine learning by learning directly from raw MRI inputs. Unlike radiomics pipelines that rely on segmentation and handcrafted feature extraction, DL architectures can automatically identify and weigh complex patterns across multiparametric sequences, offering more consistent and potentially generalisable predictions.

Cai and colleauges [46] have shown that fully automated CNNs can achieve radiologist-level accuracy (AUCs 0.86–0.89) without manual annotation, while Grad-CAMs offered interpretable tumour localisation. Johnson et al. [47] then emphasised reproducibility, releasing an open-source DL model that achieved AUCs of 0.86 for PI-RADS ≥3 and 0.78 for csPCa, highlighting the role of open science in enabling external benchmarking.

Serrano et al. [48] reported a pooled AUC of ~0.823 (95% CI 0.72–0.92) for MRI-based radiomics in PI-RADS 3 lesions, but also highlighted modest methodological quality, with a mean Radiomics Quality Score (RQS) of approximately 15 out of 36. The RQS is a standardised scoring system used to evaluate the scientific rigour, reproducibility, and clinical readiness of radiomics studies (range 0–36, with higher scores indicating stronger methodological quality and better translational potential). A mean score around 15 therefore reflects typical weaknesses in the field, including limited external validation, small datasets, and incomplete reporting [48]. Complementing this, Zhang et al. [49] meta-analysed diagnostic performance and found pooled validation-set sensitivity/specificity of 0.76/0.82 (AUC 0.77) for csPCa, improving to 0.80/0.82 (AUC 0.85) in independent validations. Together, these findings illustrate a familiar pattern: strong performance in development cohorts, some attenuation with external testing, but overall clinically meaningful discrimination.

Across single- and multicentre studies, radiomics typically delivers AUCs ~0.75–0.89 for csPCa discrimination in PI-RADS 3 lesions, with clinical–radiomic models frequently outperforming either component alone and offering net benefit on DCA in clinically relevant thresholds. Therefore, radiomics and image-based AI are promising for PI-RADS 3, but broader, well-designed external validations are essential.

## 6. Radiomics and Image-Based AI in Combination with Clinical Predictive Models

The strongest evidence favours integrated models (radiomics + PSAD/age), ideally validated across centres with clear calibration and DCA (Table 1). Combining clinical predictors with radiomics often improves robustness. Jin et al. [50] integrated four selected radiomic features with PSA and age to build a nomogram in 103 men (PI-RADS 3), achieving test AUC 0.88, outperforming radiomics-only (AUC 0.71), with favourable calibration and decision-curve analysis (DCA). In a larger two-centre study, Li et al. [51] developed a T2W/ADC/DCE-based radiomics signature and combined it with PSAD; the resulting nomogram achieved AUC 0.884 (test) and 0.907 (external validation), with good calibration and clinical utility on DCA (Table 1).

Deniffel et al. [53] compared a locally developed clinical risk model against previously published strategies in men with PI-RADS 3 lesions across two institutions. In the validation cohort, the model achieved the highest net benefit at all clinically relevant thresholds, avoiding 547 unnecessary biopsies per 1000 men at a 10% risk threshold without missing csPCa. By comparison, normalised ADC and PSA density avoided 223 and 210 biopsies, respectively, while established calculators such as the MRI-ERSPC risk model performed less well. These findings underscore that relatively simple, locally calibrated models based on clinical and imaging variables can provide tangible clinical benefit, even before incorporating more complex radiomics or DL approaches (Figure 1).

Bachetti et al. [54] embedded DL into risk calculators, analysing 538 men who underwent MRI and biopsy. Their multimodal model, combining clinical and imaging inputs, achieved the best discrimination (AUC 0.822) and reduced unnecessary biopsies by up to 43% compared with clinical-only models. Umapathy et al. [52] provided evidence of scalability, training a representation-learning model on more than 28,000 MRI examinations. Applied to PI-RADS 3 lesions, it avoided 41% of benign biopsies while maintaining sensitivity, and when combined with clinical data, nearly halved unnecessary procedures.

Together, these studies illustrate the rapid evolution of DL for PI-RADS 3 decision-making—from smaller multimodal risk calculators to large-scale representation learning, fully automated CNNs, and open-source pipelines. The consistent theme is the potential to reduce unnecessary biopsies while preserving sensitivity for csPCa, provided that broader prospective validation is pursued.

## 7. External Validation and Generalisability

A consistent limitation across the literature is the lack of reproducibility on external validation. Different institutions often identify distinct “radiomic signatures,” reflecting variability in scanners, acquisition protocols, lesion contouring, preprocessing pipelines, and underlying patient populations. As a result, single-centre models that perform well internally frequently show a marked decline when applied to external cohorts. This issue has been highlighted by several comparative studies and emphasised by Corsi et al. [55], who argue that real progress will depend on standardisation of feature computation and reporting, the creation of larger multicentre datasets spanning multiple vendors and sites, and the open sharing of code and models for independent validation.

Most models are retrospective and single-centre, raising concerns about overfitting and limited applicability. Bertelli et al. [56] demonstrated that ML/DL frameworks trained on PI-RADS v2.0 data failed to generalise when tested on v2.1 datasets, with AUROC values collapsing toward chance performance. Similarly, many radiomics models that perform well internally lose accuracy externally, underscoring the fragility of handcrafted features to acquisition variability.

Encouragingly, recent multicentre work, such as that by Jin et al. [40], has demonstrated that with harmonised pipelines and external calibration, radiomics models can achieve comparable performance across sites, suggesting that these barriers, while real, are not insurmountable.

The landmark PI-CAI study [13] in Lancet Oncology addressed this gap on a global scale, benchmarking AI against 62 radiologists across more than 10,000 MRI examinations. AI achieved a superior AUROC of 0.91 compared with radiologists (0.86), detecting 6.8% more clinically significant PCa at matched specificity while reducing false positives and overdiagnosis. Crucially, its performance was consistent across scanners, vendors, and institutions—a major advance given the variability of PI-RADS 3 interpretation. However, when benchmarked against multidisciplinary radiology reporting (the standard of care), AI did not achieve non-inferiority due to slightly lower specificity, reinforcing its role as an adjunct rather than a replacement.

Collectively, these studies demonstrate that AI can scale from single-centre feasibility to multicentre and international contexts, provided harmonised acquisition, federated learning, and open-source benchmarking are prioritised. Without these safeguards, models risk remaining academic prototypes rather than clinical tools.

## 8. Path Forward for Workflow Standardisation

The challenges highlighted by Lim et al. [38] regarding workflow variability, heterogeneous acquisition protocols, and differing radiology practices underscore a major barrier to consistent AI performance in PI-RADS 3 lesion assessment. A clear path forward requires harmonisation at several levels. First, standardised multiparametric MRI acquisition and reporting protocols are essential to reduce scanner- and site-related variability that directly affects AI reproducibility [57]. Second, training and validating models on diverse multicentre datasets—as demonstrated in recent large-scale efforts such as Umapathy et al. [52] and the international PI-CAI benchmark—can markedly improve generalisability and mitigate overfitting to single-centre conditions. Third, open-source dissemination of model weights, preprocessing pipelines, and calibration tools will allow transparent external benchmarking and accelerate refinement across institutions [13]. Together, these steps outline a pragmatic route toward standardising the AI-assisted diagnostic workflow that Lim et al. [38] identified as urgently needed.

## 9. Reducing Subjectivity and Inter-Reader Variability

While machine learning—particularly deep learning—removes the subjectivity inherent to manual lesion segmentation, its value extends further by directly addressing inter-radiologist variability in PI-RADS assessment [58]. Models trained on raw multiparametric MRI data can learn stable imaging representations that are less sensitive to individual reader interpretation, scanner differences, or site-specific practices. This results in more reproducible risk estimates than those derived from visual assessment alone [59]. Furthermore, modern deep learning systems provide calibrated probability outputs and interpretable activation maps (such as Grad-CAM), offering a consistent decision anchor that radiologists can reference [60]. When integrated into structured reporting or multidisciplinary workflows, these objective outputs help stabilise decision-making around equivocal PI-RADS 3 lesions and promote more uniform biopsy or surveillance strategies across readers and institutions. Thus, machine learning not only overcomes the subjectivity of segmentation but also supports harmonised interpretation in a domain long characterised by inter-observer variability.

## 10. Translation into Practice: Thresholds, Safety-Netting, and Shared Decisions

The value of AI in the management of PI-RADS 3 lesions ultimately depends not only on diagnostic performance but on its safe and effective integration into real clinical pathways. Successful translation requires clearly defined risk thresholds, robust safety-netting strategies, seamless workflow integration, and strong governance frameworks to ensure reproducibility, equity, and safety.

Before deployment, centres should define actionable thresholds for AI-derived risk that reflect local disease prevalence and clinical priorities [51]. Men at very low risk, such as those with PI-RADS 3 lesions and PSA density below 0.15 ng/mL/cm^3^, may safely defer biopsy with structured follow-up including repeat PSA and MRI at 6 to 12 months [51,61]. Higher-risk patients, for example, those with PSA density at or above 0.15 ng/mL/cm^3^ or rising risk estimates despite a PI-RADS 3 score, should undergo timely targeted biopsy [51].

Intermediate-risk cases warrant shared decision-making and may benefit from adjunctive tools such as biomarkers or risk calculators [62]. Clear safety nets are important: escalation should be triggered by increasing PSA density, upgrading of MRI findings, evolving symptoms, or other concerning clinical features, with defined re-imaging intervals to ensure timely reassessment [63].

## 11. Translation into Practice: Workflow Integration and Governance

For AI systems to be adopted safely in clinical practice, they must integrate smoothly into existing radiology and urology workflows, including direct compatibility with PACS/RIS platforms and structured reporting systems. Such integration is essential for reproducibility, clinician trust, and real-world usability [64]. Beyond simple accuracy metrics, AI tools must be intuitive, traceable, and clinically interpretable, with explainable outputs that highlight the imaging regions contributing to the prediction; these principles align with established reporting and transparency frameworks such as the Checklist for Artificial Intelligence in Medical Imaging (CLAIM) [65] and foundational work on explainable machine learning [66]. The importance of transparency and standardisation has further been underscored by the PI-CAI international benchmark, which demonstrated how open-source model weights, preprocessing pipelines, and evaluation frameworks enable fair comparison across institutions and promote reproducible model development [13]. Together, these elements establish the foundation for clinically deployable AI systems that support radiologist-led decision-making while meeting regulatory expectations for interpretability, robustness, and safety. Incorporating these elements creates a workflow in which AI augments, rather than replaces, expert judgement. A schematic model of this recommended workflow is presented in Figure 1, illustrating the sequential steps from MRI acquisition through AI inference, radiologist review, and multidisciplinary decision-making.

## 12. Conclusions

PI-RADS 3 lesions remain the most challenging category in prostate MRI, reflecting an intermediate and heterogeneous risk of clinically significant prostate cancer. Conventional adjuncts—such as PSA density, risk calculators, and molecular biomarkers—offer only modest incremental value and are hindered by inconsistent validation and variable uptake across clinical settings. Artificial intelligence represents a potential step-change. Across multiple studies, machine learning and deep learning models have demonstrated superior ability to differentiate benign from malignant PI-RADS 3 lesions compared with traditional parameters, reducing unnecessary biopsies while preserving sensitivity for clinically significant disease.

However, the existing evidence, while encouraging, remains early-stage. Most published models are retrospective, single-centre, and methodologically heterogeneous, with limited external or prospective validation. The most promising advancements emerge from multicentre machine learning frameworks that combine clinical, radiomic, and MRI-derived features, and from deep learning systems capable of automated, standardised image interpretation at scale. These approaches not only enhance diagnostic accuracy but also directly address one of the core limitations of mpMRI interpretation: substantial inter-reader variability, particularly within the transition zone.

## 13. Take Home Points

PI-RADS 3 lesions remain a diagnostic grey zone—common but with only modest rates of clinically significant prostate cancer—creating uncertainty in biopsy and surveillance decisions.Radiomics and image-based AI show strong promise in distinguishing indeterminate lesions by capturing subtle imaging features beyond human perception, paving the way for more objective risk stratification. However, these technologies still require large-scale, prospective validation and transparent, explainable deployment before widespread clinical use.Combining imaging data with clinical and demographic parameters—such as PSA density, age, and prostate volume—consistently enhances predictive performance over any single modality alone.

## Figures and Tables

**Figure 1 cancers-18-00028-f001:**
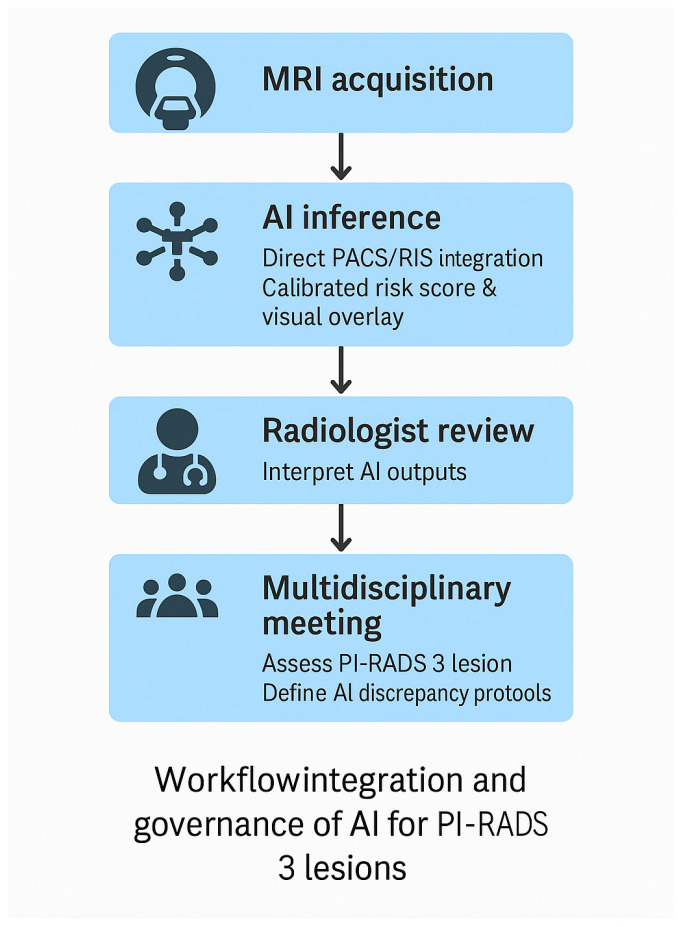
Proposed workflow for clinical integration and governance of AI tools for PI-RADS 3 lesion assessment. AI inference is embedded directly into PACS/RIS, generating calibrated risk scores and visual overlays. Radiologists retain interpretive oversight and review AI outputs. Multidisciplinary meetings adjudicate discrepancies and define escalation pathways. Governance includes model versioning, performance monitoring, drift detection, and incident reporting.

**Table 1 cancers-18-00028-t001:** Summary of major AI and radiomics studies for risk stratification of equivocal (PI-RADS 3) or indeterminate mpMRI lesions.

Study	Cohort/Design	Modality and Features	Model/Algorithm	Performance (AUC)	Key Findings/Clinical Implication
Giambelluca et al. [34]	43 pts, 46 PI-RADS 3 lesions	T2 + ADC texture features	GLM/Discriminant Analysis	0.77–0.82	Texture features improved discrimination of malignant vs. benign PI-RADS 3 lesions, supporting quantitative image analysis.
Hou et al. [35]	263 PI-RADS 3 pts (retrospective)	T2WI, DWI, ADC radiomics	Radiomics ML models (RML-i/ii)	0.89	Radiomics models outperformed PI-RADS alone, providing reproducible csPCa prediction.
Hectors et al. [36]	240 pts (train/test)	T2 radiomics (107 features)	Random Forest Classifier	0.76	ML on T2 features outperformed PSA density and prostate volume for csPCa prediction.
Brancato et al. [37]	73 lesions (41 PI-RADS 3 + 32 upgraded 4)	T2, ADC, DCE radiomics	mRMR + Bootstrap Model	0.80/0.89	T2 and ADC radiomics improved PCa detection vs. PI-RADS v2.1; DCE added limited value.
Li et al. [51]	306 train/test + 65 external	T2, ADC, DCE radiomics + PSAD	LASSO + Radiomics Nomogram	0.84–0.94	Nomogram integrating radiomics and PSAD showed excellent calibration and external validity.
Gravina et al. [41]	109 PI-RADS 3 pts	Clinical + radiologic (PSA, PSAD, BMI, volume)	Tree-based ML	NOT REPORTED	Clinical ML predicted PCa probability; PSAD and lesion location were most influential.
Jin et al. [40]	463 pts/4 centres	T2, DWI, ADC radiomics (2347 features)	SVM + ANOVA ranking	0.80 (csPCa)	Multicentre model generalised well; ADC features most robust predictors.
Zhao et al. [43]	243 TZ lesions (2 centres)	T2 + ADC radiomics + PSAD	XGBoost/Logistic Model	0.91 (val.)	XGBoost model best performance; Mean ADC + PSAD nearly equivalent—supports simpler metrics.
Altıntaş et al. [42]	235 PI-RADS 3 pts (fusion biopsy)	ADC, Ktrans, size + inflammatory indices	Random Forest	0.92	RF model identified ADC ratio and PSAD as key predictors; inflammatory indices added value.
Cai et al. [46]	5735 MRI (5215 pts) multi-site	mpMRI (T2, DWI, ADC, DCE)	CNN (Deep Learning)	0.89 (int)/0.86 (ext)	Fully automated DL model matched radiologists; Grad-CAMs localised tumours.
Johnson et al. [47]	151 bpMRI external val.	Biparametric MRI (open-source DL)	CNN (Open model)	0.86 (≥3)/0.78 (csPCa)	High sensitivity and reproducibility; first open-source external validation.
Umapathy et al. [52]	21 938 men (28 263 MRI)	bpMRI (T2 + DWI)	Representation Learning (DL)	0.73 (PI-RADS 3)/0.88 (all)	Representation learning disambiguated PI-RADS 3; avoided 41% benign biopsies.
Deniffel et al. [53]	278 PI-RADS 3 (two centres)	Clinical (ADC, PSAD, vol.)	Local Logistic Model	NOT REPORTED (net benefit)	Local model outperformed PSAD and ERSPC; reduced unnecessary biopsies.
Saha et al. (PI-CAI study) [13]	10 207 MRI (9129 pts) > 20 sites	mpMRI (PI-RADS 2.1)	Consortium AI System	0.91 (AI) vs. 0.86 (Rads)	Global confirmatory trial: AI non-inferior and superior to radiologists; 6.8% ↑ true positives, 20% ↓ false positives.

Radiomics studies (2019–2021) established feasibility (AUC ≈ 0.75–0.82). Hybrid radiomics–clinical nomograms (2022–2023) improved discrimination (AUC > 0.85). Deep learning models (2024–2025) achieved radiologist-level accuracy (AUC ≈ 0.88–0.91). Global benchmarking (PI-CAI 2024) confirmed AI’s non-inferiority to experts, supporting safe, reproducible biopsy triage in equivocal mpMRI findings. Abbreviations: AUC = area under the receiver operating characteristic curve; ADC = apparent diffusion coefficient; PSAD = prostate-specific antigen density; csPCa = clinically significant prostate cancer; DCE = dynamic contrast-enhanced; RF = random forest; CNN = convolutional neural network; SVM = support vector machine.

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
