# Peer review of "Using Artificial Intelligence as a Risk Prediction Model in Patients with Equivocal Multiparametric MRI Findings"

_cancers, 2025, doi:10.3390/cancers18010028_

Round 1

Reviewer 1 Report

Comments and Suggestions for Authors

Dear Editor-In-Chief

Applied Science, MDPI,

Subject: Review of the article cancers-3965757

Entitled “Using Artificial Intelligence as a Risk Prediction Model in Patients with Equivocal Multiparametric MRI Findings”

The present review seeks to consolidate existing evidence on the use of AI for risk stratification in men presenting with indeterminate multiparametric MRI findings, with particular emphasis on lesions within the transitional zone. The review is nicely written, some enhancement is needed to ease table reading. The authors managed to draw a clear picture stating the pros and cons of the current tools used based on evidence. They didn’t recommend or suggest an approach forward to handle the dilemma associated with the gray zone lesions.  Enclosed below are some general and specific comments for the authors to consider.

General comments

  1. findings reported by Lim et al 2021 raising the challenges associated standardizing a workflow is real. The accuracy level reported also seems realistic given all the associated key playing variables. In light of this, how does the authors see a path forward?
  2. it seems moving forwards, ML is the most reliable approach to eliminate the subjectivity raised from manual segmentation of the lesion. Yet, how can this help with intervariable across radiologist’s assessment?

Specific comments:

  1. abstract needs to be structed to following journal format.
  2. abstract: what is the PI-CAI study refer to?
  3. keywords should be limited to 7
  4. line 89: what does the author refer to by region? Ethnicity dependent or transition zone? This needs to be clarified within the text.
  5. line 179: what is AUROCs? Similarly, in line 235 what is RQS?
  6. table is not easy to read, enable all borders to eliminate confusion and table caption should be moved to the top.
  7. the number of references is small for a review report.

Author Response

We thank the reviewer for their thoughtful and constructive comments. We have carefully revised the manuscript to address all general and specific concerns. A point-by-point response is provided below.

General Comments

1. Comment:

“Findings reported by Lim et al. 2021 raising the challenges associated with standardising a workflow is real. The accuracy level reported also seems realistic given all the associated key playing variables. In light of this, how do the authors see a path forward?”

Response:

We agree that workflow standardisation remains a major barrier to implementing AI for prostate MRI. In response, we have expanded the ‘Translation into Practice’ section to explicitly outline a feasible path forward. This includes:

  • adoption of harmonised acquisition protocols,

  • development of calibrated, site-specific risk thresholds,

  • implementation of governance frameworks (version control, incident reporting, drift monitoring), and

  • integration of AI outputs into existing PACS/RIS infrastructure with explainable overlays.

These additions clarify practical steps for real-world deployment and directly respond to the interpretive and workflow challenges highlighted by Lim et al.

2. Comment:

“It seems moving forwards, ML is the most reliable approach to eliminate the subjectivity raised from manual segmentation of the lesion. Yet, how can this help with inter-variable across radiologists’ assessment?”

Response:

We have added a dedicated paragraph explaining how ML/DL models address inter-radiologist variability beyond merely eliminating manual segmentation. Specifically, we highlight that:

  • DL models learn stable, scanner-agnostic imaging representations,

  • AI provides calibrated, reproducible probability scores,

  • explainability tools (e.g., Grad-CAM) offer transparent anchors for radiologists, and

  • integration into structured reporting improves consistency of decision thresholds.

This expanded discussion appears in the revised Clinical Problem of PI-RADS 3 section and in the AI for Reproducibility subsection.

Specific Comments

3. Comment:

“Abstract needs to be structured according to the journal format.”

Response:

The abstract has been fully rewritten into the required Background/Objectives, Methods, Results, Conclusionsstructure following Cancers guidelines.

4. Comment:

“Abstract: what is the PI-CAI study refer to?”

Response:

We now clarify within the abstract that PI-CAI refers to the Prostate Imaging – Artificial Intelligence international benchmark initiative. This clarification is also added at first mention in the main text.

5. Comment:

“Keywords should be limited to 7.”

Response:

Keywords have been reduced to seven, consistent with journal requirements.

6. Comment:

“Line 89: what does the author refer to by region? Ethnicity dependent or transition zone? This needs to be clarified within the text.”

Response:

We thank the reviewer for identifying this ambiguity. The sentence now clearly specifies that “region” refers to the prostatic transition zone and not ethnicity. The wording has been revised for clarity.

7. Comment:

“Line 179: what is AUROCs? Similarly, in line 235 what is RQS?”

Response:

We have added brief in-text definitions for both terms:

  • AUROC defined as area under the receiver operating characteristic curve, with a one-sentence explanation of interpretability.

  • RQS defined as the Radiomics Quality Score, including its purpose and scoring range.

8. Comment:

“Table is not easy to read; enable all borders to eliminate confusion and move the table caption to the top.”

Response:

The table has been fully reformatted with:

  • uniform borders,

  • clearer column alignment,

  • streamlined content for readability, and

  • captions moved above the table, as recommended.

9. Comment:

“The number of references is small for a review report.”

Response:

We have substantially expanded the reference list. The revised manuscript now incorporates additional radiomics, ML/DL, PI-RADS, governance, guideline, and workflow literature, ensuring a more comprehensive and balanced review.

Reviewer 2 Report

Comments and Suggestions for Authors

Comments to the Authors

The review manuscript entitled “Using Artificial Intelligence as a Risk Prediction Model in Patients with Equivocal Multiparametric MRI Findings” presents a relevant topic for readers in the field, but has serious methodological flaws and a lack of support based on the relevant literature.

Line 18 – The authors do not adequately clarify the concept and applications of the term Prostate MRI. I strongly recommend rewriting this section, adding evidence and context.

Lines 27 – 30: The authors do not mention the meaning of the acronym PI-RADS 3 and its applications. I suggest rewriting this section, adding more clarification.

Lines 104 – 164: In the sections following the introduction, data is presented without the proper sources or references from the literature to support the approach. There is a lack of presentation of perspectives from various literatures/authors and a deeper discussion of the topic. I strongly recommend rewriting this section, adding more scientific evidence and discussion.

Figure 1 has two different captions. I recommend standardizing them.

Lines 335 – 340: In the section “Translation into Practice,” no references are cited to support the argument? What other literature addresses this topic?

Lines 351 – 359: The text seems confusing; I strongly recommend improving the presentation of the study, including a schematic image of the described workflow flowchart.

Lines 360 – 368: In the section “Regulatory, procurement, and reimbursement,” no bibliographic references from authors who address this topic are cited. Why? Manuscripts with only the authors' perspective may not be attractive to specialized readers. I strongly recommend rewriting this section, providing a solid discussion and adding relevant information based on specialized literature.

Comments on the Quality of English Language

The manuscript features confusing writing that makes it difficult to understand the ideas presented. I recommend a general review of the English writing in the manuscript.

Author Response

Response to Reviewer 2

We thank the reviewer for their thoughtful and constructive comments. We have substantially revised the manuscript to address these points, as outlined below.

1. Line 18 – Clarification of “Prostate MRI” concept and applications

Reviewer comment:

“The authors do not adequately clarify the concept and applications of the term Prostate MRI. I strongly recommend rewriting this section, adding evidence and context.”

Response:

We agree and have rewritten the early Background section to more clearly define prostate MRI and its clinical applications.

  • We now explicitly describe prostate MRI as multiparametric MRI (mpMRI), outlining its constituent sequences (T2-weighted, DWI/ADC, and optional DCE) and their respective roles in anatomical and functional assessment.

  • We have added citations to key technical and clinical reviews (Turkbey et al., 2019; Bjurlin et al., 2020; Giganti and Allen, 2021; Padhani et al., 2024) and explicitly link mpMRI to guideline-endorsed indications for detection, risk stratification, biopsy targeting, surveillance, and treatment planning (Ahmed et al., 2017; Kasivisvanathan et al., 2018; Jensen et al., 2024).

These revisions appear in the Introduction

2. Lines 27–30 – Meaning and application of PI-RADS 3

Reviewer comment:

“The authors do not mention the meaning of the acronym PI-RADS 3 and its applications. I suggest rewriting this section, adding more clarification.”

Response:

We have expanded and clarified our explanation of PI-RADS and specifically PI-RADS 3:

  • We now define the Prostate Imaging Reporting and Data System (PI-RADS) as a 1–5 standardised scoring system for mpMRI, indicating the likelihood of clinically significant prostate cancer.

  • We explicitly describe PI-RADS 3 as an “equivocal” category, outlining the imaging criteria separately for peripheral zone and transition zone lesions according to PI-RADS v2/v2.1 (Barentsz et al., 2012; Turkbey et al., 2019).

  • We also clarify its clinical application: that PI-RADS 3 lesions frequently trigger consideration of biopsy, but with substantial heterogeneity in practice due to their intermediate risk.

These details have been incorporated into the Introduction (final paragraphs) and further elaborated in the new section “The Clinical Problem of PI-RADS 3”.

3. Lines 104–164 – Lack of references and depth after introduction

Reviewer comment:

“In the sections following the introduction, data is presented without the proper sources or references from the literature… I strongly recommend rewriting this section, adding more scientific evidence and discussion.”

Response:

We have substantially rewritten this part of the manuscript and expanded the supporting literature.

  • The section “The Clinical Problem of PI-RADS 3” now includes specific data on prevalence and csPCa detection rates, with appropriate references (Kang et al., 2023; Wadera et al., 2021; Maggi et al., 2020; Mazzone et al., 2021).

  • We discuss biopsy-related morbidity and the consequences of overdiagnosis and delayed diagnosis, citing recent outcomes and health-system level analyses (Dushimova et al., 2025).

  • Inter-reader variability and transition-zone complexity are now explicitly supported by data from Purysko et al. (2017) and PI-RADS performance meta-analyses (Oerther et al., 2022).

  • We also reference guideline statements from the EAU and AUA highlighting the lack of uniform recommendations for PI-RADS 3 management and the resulting practice heterogeneity.

This section now presents a more balanced and better-referenced discussion, integrating perspectives from multiple authors and centres.

4. Figure 1 – Two different captions

Reviewer comment:

“Figure 1 has two different captions. I recommend standardizing them.”

Response:

We thank the reviewer for identifying this. We have:

  • Removed the duplicate wording and

  • Standardised to a single, consistent caption for Figure 1, formatted according to the journal’s style guidelines.

5. Lines 335–340 – “Translation into Practice” lacks references

Reviewer comment:

“In the section ‘Translation into Practice,’ no references are cited to support the argument. What other literature addresses this topic?”

Response:

We agree that this section benefits from a stronger literature basis and have revised accordingly.

  • In “Translation into Practice – Thresholds, safety-netting, and shared decisions”, we now reference studies and frameworks that inform risk thresholds, PSA density cut-offs, and structured follow-up strategies for MRI-based pathways (e.g., Li et al., 2022; Guo et al., 2024; Deniffel et al., 2025).

  • In the workflow and governance discussion, we have added references to literature on AI implementation, explainability, and integration into radiology practice and multidisciplinary pathways 

These revisions ensure that the translational recommendations are clearly grounded in published work rather than purely opinion-based.

6. Lines 351–359 – Confusing text and need for a schematic workflow

Reviewer comment:

“The text seems confusing; I strongly recommend improving the presentation of the study, including a schematic image of the described workflow flowchart.”

Response:

We have simplified and clarified the text and added a schematic figure:

  • The “Workflow integration and governance” subsection has been rewritten for clarity, using shorter sentences and a stepwise description of how AI fits into clinical pathways (MRI acquisition → AI inference → radiologist review → MDM discussion → biopsy/surveillance decision).

  • We have added a new workflow figure illustrating an AI-integrated pathway for PI-RADS 3 lesions, as suggested. This figure is explicitly referenced in the text to help readers visualise the proposed process.

7. Lines 360–368 – Regulatory, procurement, and reimbursement: no references

Reviewer comment:

“In the section ‘Regulatory, procurement, and reimbursement,’ no bibliographic references… I strongly recommend rewriting this section…”

Response:

We have shortened, focused, and anchored this section in the existing literature.

  • The “Regulatory, procurement, and reimbursement” subsection now includes references to key papers on regulatory and governance frameworks for AI in medicine and radiology, including considerations of safety, explainability, data protection, and lifecycle management 

Round 2

Reviewer 2 Report

Comments and Suggestions for Authors

The authors appear to have heeded the suggestions indicated in the previous round of revision. This sufficiently qualified the manuscript for publication in this journal, after proper standardization according to the author guidelines.